# Herb-Induced Liver Injury by Ayurvedic Ashwagandha as Assessed for Causality by the Updated RUCAM: An Emerging Cause

**DOI:** 10.3390/ph16081129

**Published:** 2023-08-10

**Authors:** Goran Bokan, Tanja Glamočanin, Zoran Mavija, Bojana Vidović, Ana Stojanović, Einar S. Björnsson, Vesna Vučić

**Affiliations:** 1Internal Medicine Clinic, Department of Gastroenterology and Hepatology, University Clinical Centre of the Republic of Srpska, 78000 Banja Luka, Republic of Srpska, Bosnia and Herzegovinazoranmavija@yahoo.com (Z.M.); 2Faculty of Medicine, University of Banja Luka, 78000 Banja Luka, Republic of Srpska, Bosnia and Herzegovina; 3Faculty of Pharmacy, University of Belgrade, 11000 Belgrade, Serbia; 4Bežanijska Kosa Clinical Hospital Center, 11000 Belgrade, Serbia; dr.anastojanovic@gmail.com; 5Landspitali University Hospital, 101 Reykjavik, Iceland; 6Faculty of Medicine, University of Iceland, 101 Reykjavik, Iceland; 7Group for Nutritional Biochemistry and Dietology, Centre of Research Excellence in Nutrition and Metabolism, Institute for Medical Research, National Institute of Republic of Serbia, University of Belgrade, 11000 Belgrade, Serbia; vesna.vucic@imi.bg.ac.rs

**Keywords:** ashwagandha, herb-induced liver injury (HILI), hepatotoxicity, liver injury, updated RUCAM

## Abstract

Herb-induced liver injury (HILI) caused by herbal supplements, natural products, and products used in traditional medicine are important for differential diagnoses in patients with acute liver injury without an obvious etiology. The root of *Withania somnifera* (L.) Dunal, commonly known as ashwagandha, has been used in Ayurvedic medicine for thousands of years to promote health and longevity. Due to various biological activities, ashwagandha and its extracts became widespread as herbal supplements on the global market. Although it is generally considered safe, there are several reported cases of ashwagandha-related liver injury, and one case ended with liver transplantation. In this paper, we review all reported cases so far. Additionally, we describe two new cases of ashwagandha hepatotoxicity. In the first case, a 36-year-old man used ashwagandha capsules (450 mg, three times daily) for 6 months before he developed nausea, pruritus, and dark-colored urine. In the second case, a 30-year-old woman developed pruritus after 45 days of using ashwagandha capsules (450 mg). In both cases, serum bilirubin and liver enzymes (aspartate transaminase (AST), alanine transaminase (ALT), and alkaline phosphatase (ALP) were increased. The liver injury pattern was hepatocellular (R-value 11.1) and mixed (R-value 2.6), respectively. The updated Roussel Uclaf Causality Assessment Method (RUCAM) (both cases with a score of seven) indicated a “probable” relationship with ashwagandha. Clinical and liver function improvements were observed after the discontinuation of ashwagandha supplement use. By increasing the data related to ashwagandha-induced liver injury, these reports support that consuming ashwagandha supplements is not without its safety concerns.

## 1. Introduction

Drug-induced liver injury (DILI) is a common cause of acute liver failure caused by drugs or other xenobiotics, which could be direct (intrinsic or idiosyncratic types) or indirect types. While intrinsic DILI is typically dose-dependent and predictable, idiosyncratic does not depend only on the dose [1]. In recent decades, there has been a rising interest in herb-induced liver injury (HILI) caused by natural products, herbal supplements, or herbal products used in traditional medicine [2]. Although they share identical or similar clinical presentations, there are differences between DILI and HILI. In contrast to DILI, which a single well-chemically characterized chemical agent causes, HILI is triggered by a chemical mixture of several constituents [3]. 

Despite the belief that all products that come from nature are safe, a recent systematic review identified 79 herbs or herbal products that induced HILI. Although most patients fully recover after discontinuing herbal products, severe hepatic traumas and fatal cases have also been reported [4]. Most of these products can be bought without a prescription. Some of these products are based on medicinal plants with a long history of use in Chinese, Ayurveda, and other traditional medicines. However, these medicinal plants and their extracts are approved for use in herbal supplements to improve physical and mental well-being, often in doses that exceed those established in traditional use [5]. Therefore, it is unsurprising that growing data in the literature indicate adverse reactions, including liver toxicity, associated with using these products [4,6,7].

Ashwagandha (*Withania somnifera*) is a plant belonging to the Solanaceae family, also known as Winter cherry or Indian ginseng. Based on phytochemical studies, *W. somnifera* contains more than 80 bioactive compounds, including steroidal lactones (withanolides) and alkaloids, as main active constituents [8]. Many studies have reported that various extracts and pure compounds from different parts of *W. somnifera* exhibited immunomodulatory, antioxidant, anti-inflammatory, anti-cancerogenic, sedative, and cardioprotective activities [9,10]. Since ancient times *W. somnifera* has been used as an anti-stress agent, aphrodisiac, for impotence and infertility treatment. In recent years, the meta-analysis of several clinical trials has supported the use of *W. somnifera* in treating male infertility [11]. Additionally, there is evidence that the oral administration of the standardized ashwagandha root extract can improve sexual function in healthy women [12]. In recent years, a growing body OF evidence has highlighted ashwagandha’s potential for stress management, cognitive function, and physical performance [13]. However, in parallel with global popularity, there are increasing concerns about ashwagandha supplements’ quality, efficacy, and safety. Although it is generally well-tolerated, several recent cases have reported ashwagandha-associated liver injury [14,15,16]. 

In addition to reporting two new cases of ashwagandha-induced liver injury that were observed in the Department of Gastroenterology and Hepatology, University Clinical Centre of the Republic of Srpska, we performed a comprehensive review of the literature on liver injury associated with ashwagandha supplements, as presented below.

## 2. Report of Cases

### 2.1. Case 1 

A 36-year-old man was hospitalized due to elevated liver enzymes, itching (without skin rash), and nausea, which started a month before hospital admission. A few days before his hospitalization, he noted a dark urine color. He denied drinking alcohol, using conventional medicaments, and having a drug addiction. His medical history revealed that he had been continuously using 450 mg capsules of ashwagandha three times a day for six previous months. He bought the capsules online and used them to improve his fertility. 

The patient’s laboratory parameters on the day of admission were total bilirubin 45 µmol/L (ref. <20), aspartate transaminase (AST) 439 IU/L (ref. <35), alanine transaminase (ALT) 1.396 IU/L (ref. <35), alkaline phosphatase (ALP) 432 IU/L (ref. 0–120), gamma-glutamyl transferase (GGT) 50 IU/L, international normalized ratio (INR) 0.98, and serum albumins 43 g/L. 

The R-value is the ratio between the values of ALT and ALP, which suggests the potential type of liver lesion, and was calculated according to the RUCAM system by dividing multiples of the upper limit of the normal range (ULN) of ALT with multiples ULN of AP. Hepatocellular injury is defined as an R-value score > 5 and cholestatic if the score is <2, whereas a mixed pattern is in between these values [17,18]. For this patient, the R-value was 11.1, indicating the hepatocellular type of liver injury.

### 2.2. Case 2

A 30-year-old woman with a skin rash and elevated liver enzymes was admitted to the hospital. A skin rash started 3 weeks before the admission. No alcohol consumption or other drugs were reported. She took 450 mg of ashwagandha capsules once daily a month and a half before the hospitalization. She used them to improve her fertility and bought them online.

Her laboratory findings showed: total bilirubin of 215 µmol/L (ref. <20), direct bilirubin of 127 µmol/L (ref. <5), AST 75 IU/L (ref. <35), ALT 111 IU/L (ref. <35), ALP 147 IU/L (ref. 0–120), GGT 11 IU/L, INR 0.93 and serum albumins 45 g/L. The R-value for this patient was 2.6, suggesting a mixed type of liver injury.

Both patients’ radiological liver examinations (abdominal ultrasound, abdominal computed tomography—CT, and magnetic resonance cholangiopancreatography—MRCP) excluded biliary obstruction, vascular obstruction, and focal liver abnormality. Based on the patient’s history, there was no likelihood of episodes of shock, hypoxia, or heart failure within 2 weeks of the onset of liver injury.

Both patients’ serology for hepatitis A virus (HAV), hepatitis E virus (HEV), hepatitis B virus (HBV), hepatitis C virus (HCV), and human immunodeficiency virus (HIV) were negative. Cytomegalovirus (CMV), Herpes simplex virus (HSV), and Epstein–Barr virus (EBV) serology were positive for IgG and negative for IgM in both patients, indicating previous but not current infections. Serology for leptospirosis in both patients was negative as well. Ceruloplasmin, alpha-1 antitrypsin, and ferritin were within the normal range in both patients, and the thyroid-stimulating hormone (TSH) and free thyroxine (FT4) were also normal. Immunological tests to exclude autoimmune hepatitis included serum antinuclear antibodies (ANA), anti-smooth muscle antibodies (ASMA), anti-liver-kidney microsome antibodies (anti-LKM 1), which were all negative, and IgG was within the normal range in both cases. 

In both cases, the ashwagandha capsules were discontinued immediately after hospitalization, and liver tests normalized within two months. In the male patient, transient liver elastography (Fibro Scan) 3 months later showed grade 2 steatosis (CAP 282 dB/m) and grade 0 fibrosis/liver stiffness of 4.3 kPa/. Transient liver elastography (Fibro Scan) in the female patient, 3 months later, showed grade 0 steatosis (CAP 226 dB/m) and grade 0 fibrosis/liver stiffness of 5.1 kPa/. 

For both patients, a diagnosis of HILI was considered very likely by excluding other competing causes of liver injury and by an improvement in liver enzyme levels two weeks after the cessation of ashwagandha capsules and complete normalization after 4 and 8 weeks. Applying the updated RUCAM score, in which both cases had a score of seven, implied a “probable” relationship with ashwagandha [18].

## 3. Review of the Published Cases

### 3.1. Methods 

A search of the literature for previously published case reports of liver injury induced by Ashwagandha was performed using PubMed/Medline, Scopus, and Web of Science for publication in English and humans. The following search terms were used: “*Withania somnifera*”, “Ashwagandha”, “Winter cherry” or “Indian ginseng”, and ‘liver injury’, ‘liver failure’, ‘DILI’, ‘hepatotoxicity’, ‘hepatocellular’ or ‘cholestasis’. The initial search yielded 121 publications. The studies not performed in humans (*n* = 68), articles not related to hepatotoxicity (*n* = 33), review articles (*n* = 10), comments (*n* = 2), and two papers in Japanese were excluded. In one case report, ashwagandha was taken together with *Garcinia cambogia,* and this paper was excluded [19]. In another one, ashwagandha was combined with dandelion tea, vitamin E, and blood builders, which included iron, vitamin B12, and folate, and this case report was also excluded [20]. Therefore, six articles with a total of 10 cases fulfilled the predetermined criteria and were included in the final analysis. The literature data on the R-value and the type of liver injury according to the updated RUCAM score are presented in Table 1.

### 3.2. Results 

Demographics and clinical features from the published reports are presented in Table 1. In the paper by Björnsson et al., the authors published a series of 5 cases, 3 from Iceland and 2 from the USA, with suspected ashwagandha-induced liver injury [14]. In another report from the United Kingdom, a case of a 39-year-old female with acute hepatitis after using ashwagandha was reported [15]. Further, Weber and Gerbes (2021) presented a case of a 40-year-old man from Germany with acute liver injury, who took 500 mg of ashwagandha extract for over a year, then switched to 450 mg of Ashwagandha for 20 days [16]. In another case report from Germany [21], a 65-year-old woman had been taking ashwagandha for 4 weeks before admission, but the dose was not reported. Unknown doses and quality of the ashwagandha supplement were also reported in papers by Lubarska et al. [22] and Suryawanshi et al. [23].

In all these cases, other causes of liver diseases were ruled out, and no cases of death have been reported so far. Only in the cases reported by Weber and Gerbes [16] and Toth et al. [21] was a liver biopsy performed. None of the patients had severe complications such as encephalopathy or coagulopathy, and the recovery started soon after the ashwagandha had been withdrawn. The only reported case that developed acute liver failure and required liver transplantation (LT) has been recently reported in the USA by Suryawanshi et al. [23]. After LT, the patient had an uneventful recovery.

Based on the reported R-value, three patients developed a mixed type of liver injury, including two cholestatic types, and three patients had a hepatocellular type of liver injury (Table 1). For two patients, the R-value was not reported [16,21]. Weber and Gerbes [16] presented a case of acute liver injury in a 40-year-old man manifested by progressive hyperbilirubinemia and an increase in liver transaminases without previous comorbidities, without specifying the type of liver injury. Toth et al. [21] reported prominent hepatocellular and canalicular cholestasis and spotty hepatocellular necrosis accompanied by multiple ceroid-laden macrophages in the perivenular parenchyma. Suryawanshi et al. [23] reported acute liver failure with an R-value > 5 that ended with LT. 

The youngest case of documented hepatotoxicity related to ashwagandha was 21 years old, and the oldest was 65. All documented cases of hepatotoxicity had a similar presentation of the disease, predominantly presenting with nausea, jaundice, and pruritus.

## 4. Discussion

The first case report of ashwagandha-induced liver injury was reported in Japan in a 20-year-old man who had taken it combined with anti-anxiety drugs in doses more than twice the recommended amount one month before hospitalization [24]. Since then, ashwagandha-induced hepatotoxicity has been reported worldwide [22]. To our knowledge, most cases of liver injury associated with ashwagandha use have been reported in Europe [14,15,16,21,22]. From 2021 there has also been an increase in the European Rapid Alert System for Food and Feed notifications concerning ashwagandha supplements, especially those sold online [25]. Chemical and microbiological contamination, the wrong dosage, and the presence of pharmacologically active ingredients are the main factors associated with the adverse health effects of dietary supplements [26]. 

This current report adds two new cases of ashwagandha-induced liver injury to the existing literature. Both patients bought the capsules online and used them to increase fertility. The daily dose of ashwagandha in the male patient from our report was similar to other male patients with documented hepatotoxicity. However, the latency period was longer, and increased liver enzyme values were higher than in the other reported cases (Table 1). After 2 weeks of discontinuing ashwagandha capsule use, liver enzymes decreased, and complete normalization was achieved within 8 weeks.

In the second case, the latency period in a 30-year-old woman taking ashwagandha capsules (450 mg/day) was 45 days. She had slightly elevated liver enzymes compared to other cases of hepatotoxicity, while her bilirubin levels were moderately increased (Table 1). This was probably caused by her delayed medical admission three weeks after the symptoms occurred, which resulted in a decline in liver enzymes, but a marked increase in bilirubin levels. A gradual decrease in liver enzymes occurred within 2 weeks after ashwagandha discontinuation and complete normalization after 4 weeks, while elevated bilirubin levels persisted for 8 weeks until full normalization.

Among the reported cases, doses of ashwagandha capsules ranged from 450 to 1350 mg daily, although in some reports, the dose was unknown (Table 1). The shortest period for the development of hepatotoxicity was 5 days. However, we reported hepatotoxicity after continued ashwagandha use for 180 days. 

Patients in all the reviewed cases had higher ALT than AST levels. The R-value ranged from 1.4 to 26.7, suggesting different types of liver injury induced by the same type of herbal products (Table 1). All patients recovered spontaneously after the discontinuation of ashwagandha use. No fatalities were reported, and only one case needed liver transplantation following ashwagandha-induced hepatotoxicity.

For newly reported cases, the diagnosis of HILI was based on a clinical, biochemical, and immunological examination, and according to the RUCAM score, it was categorized as probable. The cases had a thorough diagnostic work-up, no other reasonable cause of liver injury was identified, and spontaneous liver enzyme normalization occurred 4 and 8 weeks after the cessation of ashwagandha capsule consumption. Most of the authors did not report the RUCAM score or reported that the RUCAM score was “consistent with a possible association of ashwagandha and DILI” [23].

Despite increased evidence, the main reason and mechanisms by which ashwagandha causes liver injury are unknown. The recommended dose for *W. somnifera* root powder in Ayurvedic medicine is 3.0–6.0 g/day [27]. The risk of idiosyncratic liver injury could be associated with using ashwagandha in daily doses higher than those used in traditional medicine, even up to 12.000 mg/day [28]. In addition to large doses, there are possibilities of adulterations of commercial products by adding undeclared leaves to the root in order to increase the content of withanolides, one of the most important constituents in ashwagandha [29]. In fact, the content of withanone, the withanolide responsible for many of ashwagandha’s medicinal properties, is 19 mg/g of the dry weight of leaves and 3 mg/g dry weight of roots [30]. However, there is evidence that withanone forms adducts in DNA. It is usually detoxified by glutathione, but under limited glutathione levels, it can cause DNA damage [31]. It is suggested that ashwagandha stimulates a reduction in glutathione in cells, leading to hepatotoxicity and liver injuries in a dose-dependent manner [22]. Another reason might be that the cases of ashwagandha-associated hepatotoxicity have been underreported until recently.

In summary, ashwagandha-induced liver toxicity is probably rare, although it might be underreported. It can be presented with jaundice, so it is important to document and report well-characterized cases and warn people that ashwagandha-based products, available without prescription and online, should be taken cautiously. Increasing cases of hepatotoxicity indicate the need for a stricter assessment of quality and safety and the monitoring of ashwagandha-based supplements.

## Figures and Tables

**Table 1 pharmaceuticals-16-01129-t001:** Published cases on the liver injuries associated with ashwagandha.

Publication	Sex	Age	Dose (mg/day)	Time to HILI (Days)	Symptoms	Total Bilirubin (mg/dL)	AST IU/L	ALT IU/L	ALP IU/L	Type of Liver Injury Based on R Value	The Updated RUCAM Score
Bjornsson et al., 2020 [14]	M	24	450–1350	7	Jaundice, nausea	8.1	176	295	147	Mixed	NR
	M	45	450–1350	83	Jaundice, nausea, pruritus	4.9	226	345	159	Mixed	NR
	M	21	450–1350	16	Jaundice, abdominal pain	5.3	150	237	146	Mixed	NR
	F	62	450	5	Jaundice, nausea, abdominal pain	5.9	71	136	219	Cholestatic	NR
	F	61	500	116	Jaundice, fatigue, pruritus	5.4	95	141	168	Cholestatic	NR
Ireland et al., 2021 [15]	F	39	154	42	Jaundice right upper quadrant pain, lethargy and pruritus	9.0	NR	1514	184	Hepatocellular	NR
Weber & Gerbes. 2021 [16]	M	40	450	20	Pruritus, jaundice	25.4	NR	<300	NR	NR	NR
Tóth et al. (2023) [21]	F	65	Not reported	14	Jaundice of the skin and sclera without any abdominal pain	17.3	41	54	298	NR	NR
Lubarska et al. (2023) [22]	M	23	Unknown	>365	Pruritus, malaise, fatigue, gastrointestinal disorders, and stool discoloration	11.5	234	490	227	Hepatocellular	6
Suryawanshi et al. (2023) [23]	F	41	Unknown	40	Fatigue, general malaise, and early satiety.	10.4	2500	3400	102	Hepatocellular	6–8 (possible)
Current report	M	36	1350	180	Nausea, pruritus, dark colour of urine	2.63	439	1396	45	Hepatocellular	7
	F	30	450	45	Pruritus	12.6	75	111	147	Mixed	7

Abbreviations: M—male; F—female; ALP—alkaline phosphatase; ALT—alanine aminotransferase; AST—aspartate aminotransferase; HILI—herb induced liver injury; R—R value; RUCAM—updated Roussel Uclaf Causality Assessment Method; NR—not reported.

## Data Availability

Data sharing not applicable.

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
