# Peer review of "Herb-Induced Liver Injury by Ayurvedic Ashwagandha as Assessed for Causality by the Updated RUCAM: An Emerging Cause"

_pharmaceuticals, 2023, doi:10.3390/ph16081129_

Round 1

Reviewer 1 Report

The review of Bokan et al. described two cases of Ashwagandha-induced liver injury together with a literature review of reports of HILI. This is a high quality and potentially interesting paper for future investigations. The review is well organized and written but certain concerns should be addressed to improve the paper.

1) The authors should describe better the definition of HILI in the introduction. Is it similar to DILI?

2) In the literature, there are several publications indicating that Ashwagandha has a protective role against hepatotoxicity. How can the authors explain their observations? Is the effect dose-dependent?

3) The authors should described the different mechanisms underlying hepatotoxicity associated with Ashwagandha described in the literature.

The quality of english language is OK, but it can be improved.

Author Response

Many thanks for your time and efforts to improve our manuscript and for the opportunity to respond to all concerns. All suggestions by editors and reviewers were carefully considered and included in the revised version of the manuscript. In addition, we rephrased the title to reflect the content better, and we extended the manuscript. In the revised manuscript, the reference list accompanies these changes. We hope the revised manuscript meets the reviewers' and editors' expectations.

The responses to your comments are provided below, and changes in the manuscript are marked in yellow.

Reviewer 1:

The review of Bokan et al. described two cases of Ashwagandha-induced liver injury together with a literature review of reports of HILI. This is a high quality and potentially interesting paper for future investigations. The review is well organized and written but certain concerns should be addressed to improve the paper.

Response:  Thank you for your positive comments.

1) The authors should describe better the definition of HILI in the introduction. Is it similar to DILI?

Response:  The introduction sentences are reworded and extended to reflect the better different types and causes of liver injury.

2) In the literature, there are several publications indicating that Ashwagandha has a protective role against hepatotoxicity. How can the authors explain their observations? Is the effect dose-dependent?

Response:  The hepatoprotective effects of Ashwagandha are based on animal model studies and attributed to the antioxidant and anti-inflammatory effects of bioactive compounds, especially withanolides. The revised manuscript is complemented by discussing possible reasons for increasing hepatotoxicity concerns associated with ashwaganda-based supplements, including possible high dose-related effects.

3) The authors should describe the different mechanisms underlying hepatotoxicity associated with Ashwagandha described in the literature.

Response:  As suggested, in the Discussion in the reviewed version of the manuscript, we added information on the potential mechanisms underlying observed ashwagandha-induced hepatotoxicity.

Reviewer 2 Report

The manuscript describes two cases of liver injury following intake of a preparation containing Ashwagandha obtained via internet.

The cases are well described and the relevant analysis using the RUCAM score has been performed. In addition, an analysis of the literature was performed.

I have only two questions:

Could you explain whether the content of the product has been analyzed for hepatotoxic impurities? Was the same provider distributing the capsules? Were the patients admitted in the same time period and were they related?

Was a liver biopsy performed? And if yes, what was the result?

Author Response

Many thanks for your time and efforts to improve our manuscript and for the opportunity to respond to all concerns. All suggestions by editors and reviewers were carefully considered and included in the revised version of the manuscript. In addition, we rephrased the title to reflect the content better, and we extended the manuscript. In the revised manuscript, the reference list accompanies these changes. We hope the revised manuscript meets the reviewers' and editors' expectations.

The responses to your comments are provided below, and changes in the manuscript are marked in yellow.

Reviewer 2:

The manuscript describes two cases of liver injury following intake of a preparation containing Ashwagandha obtained via internet. The cases are well described and the relevant analysis using the RUCAM score has been performed. In addition, an analysis of the literature was performed.

Response:  Thank you for your positive comments.

1) Could you explain whether the content of the product has been analyzed for hepatotoxic impurities? Was the same provider distributing the capsules? Were the patients admitted in the same time period and were they related?

Response:  The patients have not been admitted at the same time; they are two cases hospitalized independently.  However, both cases of hepatotoxicity were related to the same product but with different self-defined dose regimens, bought via the Internet and used for improving fertility. Unfortunately, additional chemical screening analyses of products were not performed due absence of equipment necessary for doing that at the local institutional laboratories. Also, it was not possible to send them to analyses in other countries during that period. Additionally, the product is not approved for free sale based on the national legislation requirements. Therefore it is sold illegally, via the Internet, without any previous safety control and advice from health care professionals. Risks related to self-medication practice and buying health products via the Internet are highlighted in the revised manuscript.

2) Was a liver biopsy performed? And if yes, what was the result?

Response:  A liver biopsy has not been performed. One of the reasons is that LTs resolved spontaneously over time, and the patients did not have any severe manifestation of hepatotoxicity.

Reviewer 3 Report

These case reports confirm previous publications, and new aspects were not provided. There is also a major methodology problem.

Major points:

1. The use of RECAM to assess causality is not acceptable, as this method lacks proper validation, considers DILIN cases with a method not validated as based on subjective opinion, and finally, includes data of the LiverTox database, highly debated by others and with well described shortcomings also by Björnsson and Hoofnagle.

2. The use of reference 15 is outdated for various reasons like lack of HEV infection exclusion.

3. Ref 15 must be replaced by the updated RUCAM of 2016, and this paper should correctly be referenced. 

4. Table 1 should provide published RUCAM scores.

5. Delete the note on weaknesses of RUCAM, because advantages are much more evident. Consider and quote that RUCAM outperforms any other method in terms of case numbers. 

6. RUCAM but not RECAM is in worldwide use, discuss that and provide references.

7. Throughout the text, HILI should refer to herb-induced liver injury, replacing herbal-induced liver injury. 

8. You are in error classifying the RUCAM R-value as a causality assessment method. Correct that or provide evidence in the text.

9. Include in the abstract scores provided by the updated RUCAM.

10. Key words, delete DILI, as you are not dealing with a convention drug, and include: Updated RUCAM.

Author Response

Many thanks for your time and efforts to improve our manuscript and for the opportunity to respond to all concerns. All suggestions by editors and reviewers were carefully considered and included in the revised version of the manuscript. In addition, we rephrased the title to reflect the content better, and we extended the manuscript. In the revised manuscript, the reference list accompanies these changes. We hope the revised manuscript meets the reviewers' and editors' expectations.

The responses to your comments are provided below, and changes in the manuscript are marked in yellow.

Reviewer 3:

These case reports confirm previous publications, and new aspects were not provided. There is also a major methodology problem.

Response:  We appreciate your criticism and all the constructive suggestions. We carefully revised the manuscript to address the significance of provided data better. Additionally, we reviewed, to the best of our knowledge, all available literature data related to ashwagandha-induced liver injury. We hope that we have answered all your questions in the revised manuscript.

1) The use of RECAM to assess causality is not acceptable, as this method lacks proper validation, considers DILIN cases with a method not validated as based on subjective opinion, and finally, includes data of the LiverTox database, highly debated by others and with well described shortcomings also by Björnsson and Hoofnagle.

Response: Thank you very much for these suggestions. We have removed the RECAM scoring in the revised version of the manuscript.

2) The use of reference 15 is outdated for various reasons like lack of HEV infection exclusion.

Response: Thank you very much for this observation. This reference was replaced with a relevant one.

3) Ref 15 must be replaced by the updated RUCAM of 2016, and this paper should correctly be referenced. 

Response: The revised version of the manuscript cite:  Danan, G.; Teschke, R. RUCAM in Drug and Herb Induced Liver Injury: The Update. Int. J. Mol. Sci. 2016, 17, 14.

4) Table 1 should provide published RUCAM scores.

Response: Based on your suggestion, we replaced the R-values calculated by the authors of this manuscript with original literature data and published data on the type of liver injury according to the R ratio.

5) Delete the note on weaknesses of RUCAM, because advantages are much more evident. Consider and quote that RUCAM outperforms any other method in terms of case numbers. 

6) RUCAM but not RECAM is in worldwide use, discuss that and provide references.

Response: We deleted contents that refer to RECAM throughout the whole manuscript.

7) Throughout the text, HILI should refer to herb-induced liver injury, replacing herbal-induced liver injury. 

Response: Thank you very much for this observation. Throughout the text, herbal-induced liver injury is replaced with herb-induced liver injury.

8) You are in error classifying the RUCAM R-value as a causality assessment method. Correct that or provide evidence in the text.

Response: We absolutely agree with your criticism and apologize for the misinterpretation. It was corrected in the revised version of the manuscript.

9) Include in the abstract scores provided by the updated RUCAM.

Response: As suggested, we added the updated RUCAM scores in the Abstract in the revised version.

10) Key words, delete DILI, as you are not dealing with a convention drug, and include: Updated RUCAM.

Response: The required corrections were made in the revised manuscript.

Round 2

Reviewer 3 Report

Thank you for substantial improvement. A few suggestions should be considered.

Minor points:

1. Title is confusing and should be more promotional. Please change to: Herb-induced liver injury by the Ayurvedic Ashwagandha as assessed for causality by the updated RUCAM: an emerging cause.

2. Abstract, lower part both scores of 7, change to: both cases with a sore of 7, and correct “probably” to probable, also elsewhere throughout your text.

3. Throughout the text, change R-ratio to R value. Bottom of section 2.1 regarding your ref 17, replace by the original RUCAM of 1993 and your ref 18.

4. Table 1, include RUCAM score for each case, if available or mention NR, delete R ratio and the listed values.

5. Middle of discussion, replace continual by continued.

Author Response

Reviewer 3:

Thank you for substantial improvement. A few suggestions should be considered.

Response: Once again, we are very thankful to you for your time and your constructive suggestions. Additional corrections have been made and are marked in yellow in the revised manuscript.

Minor points:

  1. Title is confusing and should be more promotional. Please change to: Herb-induced liver injury by the Ayurvedic Ashwagandha as assessed for causality by the updated RUCAM: an emerging cause.

Response: We accepted your suggestion and have changed the title of the manuscript.

  1. Abstract, lower part both scores of 7, change to: both cases with a sore of 7, and correct “probably” to probable, also elsewhere throughout your text.

Response: All corrections are done in the revised version of the manuscript.

  1. Throughout the text, change R-ratio to R value. Bottom of section 2.1 regarding your ref 17, replace by the original RUCAM of 1993 and your ref 18.

Response: Thank you very much for this observation. We have changed R-ratio to R value through the text. And, also we have replaced the ref 17 with the original RUCAM of 1993.

  1. Table 1, include RUCAM score for each case, if available or mention NR, delete R ratio and the listed values.

Response: Thank you for your suggestion. We agree that it is in line with the corrected title of the manuscript too. Although there is limited literature, instead of R values we inserted a column with RUCAM scores.

  1. Middle of discussion, replace continual by continued.

Response: We have replaced continual by continued in the discussion.
